# Proteomic Analysis of Alfalfa (*Medicago sativa* L.) Roots in Response to *Rhizobium* Nodulation and Salt Stress

**DOI:** 10.3390/genes13112004

**Published:** 2022-11-02

**Authors:** Yafang Wang, Pan Zhang, Le Li, Danning Li, Zheng Liang, Yuman Cao, Tianming Hu, Peizhi Yang

**Affiliations:** 1College of Grassland Agriculture, Northwest A&F University, Yangling, Xianyang 712100, China; 2Department of Grassland Science, College of Animal Science and Technology, Northeast Agricultural University, Harbin 150030, China

**Keywords:** ion homeostasis, proteomics, plant-microbe interaction, root nodules, symbiotic nitrogen fixation

## Abstract

(1) Background: Alfalfa is an important legume forage throughout the world. Although alfalfa is considered moderately tolerant to salinity, its production and nitrogen-fixing activity are greatly limited by salt stress. (2) Methods: We examined the physiological changes and proteomic profiles of alfalfa with active nodules (NA) and without nodules (NN) under NaCl treatment. (3) Results: Our data suggested that NA roots showed upregulation of the pathways of abiotic and biotic stress responses (e.g., heat shock proteins and pathogenesis-related proteins), antioxidant enzyme synthesis, protein synthesis and degradation, cell wall degradation and modification, acid phosphatases, and porin transport when compared with NN plants under salt stress conditions. NA roots also upregulated the processes or proteins of lipid metabolism, heat shock proteins, protein degradation and folding, and cell cytoskeleton, downregulated the DNA and protein synthesis process, and vacuolar H^+^-ATPase proteins under salt stress. Besides, NA roots displayed a net H^+^ influx and low level of K^+^ efflux under salt stress, which may enhance the salt tolerance of NA plants. (4) Conclusions: The *rhizobium* symbiosis conferred the host plant salt tolerance by regulating a series of physiological processes to enhance stress response, improve antioxidant ability and energy use efficiency, and maintain ion homeostasis.

## 1. Introduction

Leguminous plants can establish a symbiotic interaction with rhizobia and form nitrogen-fixing nodules. Within the nodules, bacteria can convert nitrogen gas (N_2_) to ammonia ions which can be used by the host [1]. The symbiotic nitrogen fixation (SNF) of legumes can survive in the area with limited nitrogen supply and alleviates the usage of synthetic fertilizers in agronomic systems [2,3]. Besides, legume plants contain high proteins, and they are an important protein source for human and animal consumption [4]. Therefore, the improvement of the SNF plant’s adaption to various environments is important in agriculture and the ecosystem.

The SNF interaction benefits both plants and bacteria. In this symbiotic relationship, many plant growth-promoting molecules and enzymes are synthesized and released, such as flavonoids, various phytohormones, lipo-Chito-oligosaccharide Nod factors (NFs), and the enzyme 1-aminocyclopropane-1-carboxylate (ACC) deaminase that can directly or indirectly stimulate plant growth under different environmental conditions [5]. Flavonoids, with anti-microbial activity, have been shown to play a significant role in plant defense response [6]. NFs have been known to modulate plant innate defense [7]. Transcriptomic data suggested NFs regulate genes that participated in plant immune response [8]. Studies showed that inoculated soybean plants contained some volatile compounds, such as menthyl acetate, linalyl acetate, and α-farnesene, which showed significantly higher antioxidant activity than uninoculated plants [9]. ACC deaminase prevents excessive ethylene synthesis in higher plants under various stress conditions, leading to increased plant productivity. It has been reported that lumichrome and riboflavin can maximize plant growth and improve plant immune response [5]. Phytohormones released by rhizobia, especially indole acetic acid (IAA), altered the rhizosphere chemistry, and promoted plant growth and productivity [10,11,12]. Several studies have shown that nodulated legumes are more tolerant to adverse environmental conditions than plants receiving mineral nitrogen [13,14,15]. Therefore, biological N fixation might be a feasible approach to increase the legumes’ performance under different environmental constraints.

Alfalfa (*Medicago sativa* L.) is an extensively cultivated perennial legume forage because of its high protein content and biomass production. Alfalfa is regarded as moderately tolerant to salinity [16]. However, its growth and productivity are greatly reduced by severe salt stress. Our previous research showed that alfalfa plants with active nodules (NA) presented a higher survival rate, probably due to the higher activities of antioxidative enzymes under NaCl stress than plants without nodules (NN) [17]. The proteomics data of alfalfa shoots show that NA plants maintained a higher level of plant defense against pathogens, photosynthesis, carbon fixation, reactive oxygen species scavenging, anion transport, and cell wall remodeling [18]. We hypothesize that *rhizobium* symbiosis changed the molecular and physiological processes that improved the salt tolerance of the host plants. While roots are essential organs that provide the plant with water and nutrients, salt stress impact is more pronounced in roots than in leaves. Therefore, we analyzed the proteomic profile of alfalfa roots with or without inoculation of rhizobia under salt stress. We also examined the antioxidant enzymatic activities, osmotic solute content, and ion flux to understand plant response to nodulation and salt stress.

## 2. Materials and Methods

### 2.1. Plant Materials and Growth Conditions

Alfalfa (*Medicago sativa* L. cv. Ladak^+^) seeds were purchased from the Clover Group of China. Seed sterilization was performed following three-step procedures: soaked in 70% ethanol for 30 s, immersed in 5% sodium hypochlorite (NaOCl) for 5 min, and washed with sterile water five times. Seeds were then placed in Petri dishes and germinated in a growth chamber with a controlled temperature of 25/15 °C (day/night) and relative humidity of 55/70% (day/night) for five days. Seedlings with uniform growth performance were transplanted to plastic cones (9 by 30 cm) prepacked with sterilized quartz sand. After transplanting, all the seedlings in the cones were then moved to the greenhouse with an approximate 25 ± 3/20 ± 2 °C (day/night) temperature and 55 ± 5/70 ± 5% (day/night) relative humidity. The seedlings were watered daily with a complete nutrition solution before the *rhizobia* inoculation and salt treatments for 60 days.

### 2.2. Rhizobia Inoculation

Alfalfa plants were randomly divided into two groups. One group was inoculated with *Rhizobium meliloti* (strain Dormal) and watered with a nitrogen (N)-free nutrient solution daily. Those plants established a symbiotic relationship with *rhizobium* and formed active nodules (NA). For the other group, plants without *rhizobia* inoculation were watered with a complete nutrient solution and did not develop any nodules (NN). The *rhizobia* inoculation treatment was the same as our previous studies and was described in detail in our previous publication [17].

### 2.3. Salt Treatments

Thirty days after inoculation, alfalfa plants of both groups were subjected to salt stress treatment. NA plants were daily watered with an N-free nutrient solution containing sodium chloride (NaCl, 150 mM), and NN plants were irrigated daily with a complete nutrient solution containing NaCl (150 mM). The two groups irrigated with respective nutrient solutions without NaCl were set as the control (normal growth conditions). The root tissues were harvested at 0 d, 5 d, 10 d, and 15 d post NaCl treatment. The roots were frozen in liquid N_2_ and stored at −80 °C for further analysis.

### 2.4. Physiological Measurements

Proline content was determined with a spectrophotometer according to the method of Bates, et al. [19]. Soluble sugar content was measured with the anthrone-sulfuric acid method [20]. The superoxide dismutase activity (SOD, EC 1.15.1.1) was determined by the nitroblue tetrazolium (NBT) method [21] and the catalase activity (CAT, EC 1.11.1.6) was measured with the method described by Raza et al. [22]. Three plants from each treatment were used for measurement in each experiment. The experiment was repeated five times. Three plants were used for measurement in each experiment. Data analysis was conducted with the independent *t*-test (α = 0.05) in SPSS 20.0.

### 2.5. Proteomic Analysis

The processes of protein extraction, quantification, iTRAQ labeling, strong cation exchange (SCX) chromatography, and LC-ESI-MS/MS identification were described in our previous publication [18] in detail. Protein identification and quantification were conducted with the Mascot software (version 2.3.02, Matrix Science). Sequence searches were blasted against the *Medicago truncatula* protein database [https://data.jgi.doe.gov/refine-download/phytozome?organism=Mtruncatula&expanded=285 (accessed on 30 October 2022)]. The criteria of false discovery rate (FDR) ≤ 0.01 was used to identify peptides and proteins. The proteins with a significant change (*p* < 0.05 and >1.2-fold or <0.83-fold change) were defined as differentially expressed proteins (DEPs) and listed in Appendix A. MapMan software (version 3.5.1) was used to perform pathways enrichment analysis with DEPs [23].

### 2.6. Net K^+^ and H^+^ Flux Measurements

The net H^+^ and K^+^ ion flux was determined with the NMT system (BIO-001A, Younger USA Science &Technology Corporation, Amherst, MA, USA). The NMT system measured the ion concentration gradients through a pre-set excursion distance (30 μm) at a frequency of 0.3 to 0.5 Hz. Prior to flux measurements, ion selective electrode calibration was conducted in a calibration solution (0.05 mM, 0.1 mM and 0.5 mM KCl for K^+^ calibration and pH 6.0, 6.5 and 7.0 for H^+^ calibration). Ion flux was calculated by Fick’s law of diffusion:J_0_ = −D(dc/dx)
in which J_0_ is the ion flux, D is the ion diffusion constant in a particular medium and dc/dx is the ion concentration gradient.

Alfalfa roots with or without nodules were carefully cleaned with distilled water and then placed in the test solution for 30 min before measurement. The test solution for H^+^ contained 0.1 mM KCl, 0.1 mM CaCl_2_, 0.1 mM MgCl_2_, 0.5 mM NaCl, 0.2 mM Na_2_SO_4_, 0.3 mM MES (pH 6.5). The test solution for K^+^ included 0.1 mM KCl, 0.1 mM CaCl_2_, 0.1 mM MgCl_2_, 0.5 mM NaCl, 0.2 mM Na_2_SO_4_, and 0.3 mM MES (pH 6.5). Alfalfa roots were then transferred to a fresh test solution for 5 min. After that NaCl was added to the test solution to reach a final concentration of 150 mM (pH 6.5). The ion flux was monitored at 1, 2, 4, 6, and 12 h of NaCl stress. The real-time ion flux was measured for 5 min when ion flux was steady for each time point.

The net flux rate was calculated with the software of imFluxes V2.0 (YoungerUSA LLC, Amherst, MA 01002, USA). The positive and negative values indicate external and internal ion flux, respectively.

## 3. Results

### 3.1. Physiological Analysis

To explore the effect of inoculation on alfalfa response to salt stress, we measured the activities of catalase (CAT) and superoxide dismutase (SOD), and the contents of proline and soluble sugar in NA and NN plant roots under 150 mM NaCl stress for different times (Figure 1). The CAT and SOD activities were higher in NA roots (*p* < 0.05) than that in NN under normal growth conditions (Figure 1A,B). Even though CAT activity declined in both NA and NN roots from 5 to 10 days of NaCl stress, NA roots displayed higher (*p* < 0.05) CAT activity under salt stress conditions compared with NN roots. NA roots also displayed higher (*p* < 0.05) SOD activity under salinity treatment compared to NN plants. While NA plants displayed higher soluble sugar content (*p* < 0.05) in roots than NN under normal and salt stress conditions (Figure 1D), the proline content in NA roots was less (*p* < 0.05) than that in NN plants (Figure 1C). We then examined proteomic changes in NA and NN roots under normal growth conditions and salt stress for 10 days.

### 3.2. Summary of Protein Identification

Based on iTRAQ quantification proteomic data, a total of 425,339 spectrums were generated, and 9759 peptides and 3713 proteins were identified with 1% FDR. The coefficient of variation (CV) value is an indicator of the repeatability of iTRAQ-based proteomics. The mean CV value is 0.19, 0.17, 0.15, and 0.12 for the comparison of CK_NA vs. CK_NN, S_NA vs. S_NN, S_NA vs. CK_NA, and S_NN vs. CK_NN, respectively (Appendix A). The protein frequency of less than 20% is 66% (CK_NA vs. CK_NN), 72% (S_NA vs. S_NN), 75% (S_NA vs. CK_NA), and 81% (S_NN vs. CK_NN). These results suggested that the proteomics data showed relatively high repeatability.

### 3.3. Differentially Expressed Proteins and Pathway Analysis of Alfalfa with or without Nodules

We identified a total of 922 DEPs in the comparison of NA and NN roots under normal growth conditions and salt stress (Appendix A). Among the 922 DEPs, 519 proteins were upregulated, and 406 proteins were downregulated in NA roots. The number of specifically expressed proteins was 154 upregulated and 138 downregulated for control conditions, 109 upregulated and 74 downregulated for salt stress; 256 proteins were upregulated and 194 were downregulated in NA roots under both control and salt stress conditions (Figure 2A,B).

To understand how alfalfa roots respond to *rhizobia* inoculation, we used Mapman to compare the protein changes among NA and NN roots. Under normal conditions, 35 pathways were upregulated in NA plants, 26 of which were over-represented (Figure 2C). The upregulated pathways were “oxidative pentose phosphate (OPP)”, “cell wall”, “stress, biotic and abiotic”, “glutathione S transferases”, “β-1,3-glucan hydrolases”, “peroxidases”, “GDSL-motif lipase”, “protein degradation”, “protein synthesis, ribosomal protein”, “DNA synthesis/chromatin structure”, “signaling, receptor kinases”, and “development, storage proteins”. There were 20 downregulated pathways in NA roots, and among them 14 pathways were over-represented. They included “major carbohydrates (CHO) metabolism”, “lipid metabolism”, “hormone metabolism, ethylene”, “redox, heme”, “RNA processing”, “DNA repair”, “DNA repair”, “protein synthesis, initiation and post-translational modification”, “cytoskeleton”, and “vesicle transport”. Under salt stress conditions, 21 of the total 30 upregulated pathways were over-represented in NA roots (Figure 2D). These processes are “cell wall”, “stress, biotic and abiotic”, “β-1,3-glucan hydrolases”, “glutathione S transferases”, “peroxidases”, “acid and other phosphatases”, “GDSL-motif lipase”, “protein synthesis, 40S subunit”, “protein degradation”, and “transport porins”. The nie over-represented pathways among the 10 total downregulated ones included “lipid metabolism”, “hormone metabolism, ethylene”, “redox, heme”, “redox, dismutases and catalases”, “oxidases”, “plastocyanin-like”, “protein synthesis, 60S subunit”, and “vesicle transport”. The proteins involved in these processes were listed in Appendix A. The common pathways shared by alfalfa roots under normal and salt stress conditions were summarized in Appendix A.

### 3.4. Differentially Expressed Proteins and Pathways Analysis of Alfalfa under Salt Stress

We identified a total of 679 DEPs in comparison to salt stress and normal growth conditions in NA and NN roots, among which 392 were upregulated and 307 were downregulated proteins (Appendix A and Figure 3A,B). NA and NN roots shared 39 upregulated and 55 downregulated proteins. We identified 131 upregulated and 178 downregulated proteins specifically in NA roots, and 222 and 74 proteins upregulated and downregulated, respectively, for NN roots.

In NN roots, the upregulated proteins under salt stress participated in “OPP”, “secondary metabolism, flavonoids”, “amino acid metabolism”, “co-factor and vitamin metabolism”, “biotic stress, pathogenesis-related (PR) proteins”, “protein, degradation, ubiquitin”, “signaling, 14-3-3 proteins”, and “development, storage proteins”. The downregulated processes in NN roots included “hormone metabolism, jasmonate”, “gluco-, galacto- and mannosidases”, “myrosinases-lectin-jacalin”, “protein degradation, subtilases”, and “transport, porins” (Figure 3C). Based on the over-represented pathways, NA roots showed nine upregulated processes under salinity, which were “lipid metabolism”, “abiotic stress”, “protein degradation and folding”, and “cell organization, cytoskeleton”. The downregulated proteins in NA roots under salt stress were involved in “DNA synthesis” and “transport, p- and v-ATPases”. However, the proteins regulating “protein synthesis” showed a complex change for some ribosomal proteins were upregulated while others were downregulated (Figure 3D). The proteins involved in these processes were listed in Appendix A.

### 3.5. K^+^ and H^+^ Ion Flux

To explore the effect of nodulation on ion balance under salt stress, we measured the K^+^ and H^+^ ion flux in the apical root tip before and after plants were subjected to 150 mM NaCl treatment.

NA roots showed a net H^+^ influx, while NN roots displayed a net H^+^ efflux under both normal growth conditions and NaCl stress (Figure 4A). Salt stress induced H^+^ flux fluctuations in NN roots and reduced the net H^+^ efflux at 6 and 12 h of salt stress. The speed of H^+^ influx was reduced after salt stress for 1, 2, and 4 h of salt stress in NA roots, and then increased to a comparable level of the normal growth conditions (−0.51 pmol cm^−2^ s^−1^) at 6 and 12 h of salt stress.

NA and NN roots displayed a net K^+^ efflux under both normal conditions and salt stress (Figure 4B). The K^+^ efflux rate (46 pmol cm^−2^ s^−1^) in NA was higher than that in NN roots (30 pmol cm^−2^ s^−1^) under normal growth conditions. Salt stress increased K^+^ efflux in both NA and NN roots to (about 210 pmol cm^−2^ s^−1^) at 1 h. while NA roots decreased the K^+^ efflux rate from 2 to 6 h of salt stress to 20 pmol cm^−2^ s^−1^ and then showed a comparable level of the normal growth conditions, NN roots showed a significantly higher rate of K^+^ efflux from 2 to 12 h of salt stress and peaked at 2 h at the level of 411 pmol cm^−2^ s^−1^.

## 4. Discussion

The physiological analysis suggested that nodulation increased the activities of antioxidant enzyme activities and improved soluble sugar content in alfalfa roots under normal growth conditions. The increased capacity of antioxidant defense and osmotic adjustment might be associated with a high level of reactive oxygen species (ROS) triggered by *rhizobium* symbiosis [17]. The activities of CAT and SOD in alfalfa root were different from those in the shoots, where a high level of antioxidant enzyme activity was observed in NA plants [18] under salt stress. These results implied that alfalfa root was more sensitive to salt stress than shoots. NA plants showed a better osmotic adjustment with a continuously higher level of soluble sugar content under different times of salt stress, compared with NN roots. We further analyzed the proteomic changes of NA and NN roots on day 10 of salt stress.

### 4.1. Nodulation Triggered Pathways of the Stress Response, Antioxidant Enzymes, Energy-Saving, and Signaling Processes

A total of 410 proteins were upregulated in the alfalfa root in response to *rhizobium* symbiosis. These upregulated proteins were involved in the processes of OPP, cell wall degradation, biotic and abiotic stress, antioxidant enzymes, protein degradation, receptor kinase signaling, and storage proteins. Whereas there were 365 downregulated proteins, participating in the processes of major CHO synthesis, fatty acid (FA) synthesis, ethylene response, heme redox, RNA processing, protein post-translational modification, cell organization, and vesicle transport.

#### 4.1.1. Stress Response

The proteins that participated in OPP were 6-phosphogluconolactonase (6PGL), ribose-5-phosphate isomerase A (RPI), and root-type ferredoxin reductase-like NAD(P) binding domain protein (RFRN). 6PGL catalyzes the hydrolysis of 6-phosphogluconolactone in the second step of the pentose phosphate pathway (PPP) [24]. Arabidopsis *pgl3* mutant altered the cellular redox state and played an important role in pathogen defense [25]. RPI is an enzyme and participates in the pentose-phosphate pathway. It has been reported that RPI played an essential role in carbohydrate metabolism and defense response to bacteria [26]. RFNR isoforms catalyze the non-photosynthetic reduction of ferredoxin and are essential for plant survival under low temperatures and pathogen infection [27]. The upregulation of OPP proteins might be essential to deal with a higher level of ROS induced by *rhizobium* symbiosis [17] and play a role in the maintenance of cellular redox homeostasis in alfalfa roots. 

Four upregulated proteins involved in cell wall degradation were polygalacturonase inhibitors (PGIPs). As a cell wall-binding protein, PGIP specifically inhibits the plant cell wall degrading enzyme and prevents pathogen invasion. It has been reported that both biotic and abiotic stress can induce the expression of *PGIP*. Therefore, PGIP plays an essential role in plant defense response [28].

There were 21 biotic response proteins upregulated by nodulation in alfalfa roots, including 14 PR proteins. The PR proteins were chitinases, protease inhibitors, thaumatin family proteins, lipid transfer proteins, LRR receptor-like kinases, cysteine-rich secretory proteins, and lipocalin-like domain proteins. PR proteins play an indispensable role in the innate immune responses of plants [29]. Chitinases defend fungi infection and allow symbiotic interaction with nitrogen-fixing bacteria by reducing the defense reaction of the plant [30]. Protease inhibitors enhanced the host plant’s defense capacity by reducing the fungal lytic enzyme activity, disrupting the replication cycles, or limiting the release of amino acids from host plants [31]. Here, the upregulation of chitinase proteins, protease inhibitors, and other PR proteins might be involved in protecting the symbiotic rhizobia from other bacteria or fungi invasion and thus improving the defense response in alfalfa roots. Another family of PR proteins was β-1,3-glucanases, which were classified as “misc, β-1,3-glucan hydrolases” in MapMan. β-1,3-glucanase plays as role in degradation of fungus cell wall β-1,3-glucans. The expression of β-1,3-glucanase was significantly increased during the infection process [32]. Therefore, nodulation induced pathogen resistance by the upregulation of PR proteins. Together with the biotic response, proteins participating in the abiotic response were also upregulated by *rhizobium* symbiosis.

Two ethylene-responsive transcriptional coactivator proteins were downregulated. Looking closely, they were multiprotein bridging factor 1 proteins (MBF1). Overexpression of *AtMBF1a* increased *Arabidopsis* plants’ tolerance to pathogens infection and salt stress [33]. The increased abundance of MBF1 may enhance alfalfa defense against biotic stress.

#### 4.1.2. Antioxidant Enzymes

There were nine upregulated proteins involved in glutathione S transferases (GSTs). GSTs mediate phase II detoxification and certain GSTs exhibit peroxidase activity [34]. Peroxidases catalyze hydrogen peroxide (H_2_O_2_) and alleviate oxidative stress. Thirty-seven peroxidases were upregulated in NA plants and the majority of them were class III peroxidases (Prxs), which are involved in plant defense, cell wall lignification, and cell elongation [35]. However, the POD activity was not significantly altered with the physiological analysis (data not shown), while other antioxidant enzyme activities, such as CAT and SOD, were increased by *rhizobium* biosynthesis.

#### 4.1.3. Energy Consumption

Proteins involved in energy metabolism were greatly changed by *rhizobium* symbiosis. The pathway of starch biosynthesis was downregulated by decreasing the protein abundance of ADP glucose pyrophosphorylase (AGPase) [36]. Acetyl-CoA carboxylase (ACCase) is a key enzyme that limited the rate of FA synthesis [37]. Two ACCase proteins were downregulated, implying that the FA synthesis was limited in plant roots when in symbiosis with *rhizobium*. The protein initiation involved in protein synthesis was downregulated while the protein degradation was upregulated. Inoculated plants might decrease protein synthesis and enhance protein degradation as a way to save energy. Therefore, plants in symbiosis with *rhizobium* might decrease energy consumption and supply metabolic resources to exchange nitrogen with bacteria.

Four storage proteins that act as nutrient reservoirs were also upregulated by *rhizobium* symbiosis, and three of them were patatin-like phospholipases (PLP). PLP are enzymes involved in lipid hydrolysis. Studies showed that this protein family has a role in regulating plant growth and development, and response to adverse environmental stress and pathogenic bacteria infection in plants [38].

#### 4.1.4. Signaling

As the largest subfamily of transmembrane receptor-like kinases, plant leucine-rich repeat receptor kinases (LRR-RKs) play crucial roles in hormone signaling, immune resistance, and abiotic stress responses [39]. Here, we identified 10 upregulated receptor kinases in alfalfa roots with nodules and among which seven proteins were LRR-RKs. Those upregulated LRR-RKs might participate in signaling transport in the legume-rhizobium symbiosis.

#### 4.1.5. Cell Cytoskeleton

The cytoskeleton creates an internal architecture throughout the cytoplasm in cells. It helps cells maintain their shape, provides mechanical support, and is responsible for the movement of the cell itself and various organelles. Cell cytoskeleton was modified by *rhizobium* symbiosis in alfalfa roots. As the largest type of filament, microtubules are major components of the cytoskeleton, and they are composed of tubulin protein [40]. Five microtubule proteins were upregulated in NA roots. Four of them were tubulin β-1 chain proteins, the major constituent of microtubules, and one was microtubule-associated proteins MAP65/ASE1, which enhances microtubule polymerization, promotes nucleation, and stabilizes microtubules against cold treatment in *Arabidopsis* [41]. Actin filaments are the smallest type of cytoskeleton, and they are made of actin protein. The actin filament bundling proteins were downregulated in NA roots compared with NN ones. The downregulation of the actin-binding proteins was companied by the upregulation of the actin depolymerizing factor (ADF) protein. ADF/cofilin is the main protein family promoting actin dynamics by depolymerizing and severing actin filaments [42]. The dynamics of the actin cytoskeleton are essential for membrane deformation and uptake of external substances or probing the environment [43]. Therefore, *rhizobium* symbiosis changed cell cytoskeleton composition by upregulation of the microtubule proteins and enhancing their stabilization, downregulation of actin filaments and promoting their depolymerization. The modification of cell organization might facilitate *rhizobium* infection or the maintenance of a symbiotic relationship.

#### 4.1.6. Protein Synthesis and Degradation

Ribosomes are the large complexes of protein-RNA in all cells. The macromolecular assemblies function as machines for protein synthesis. Ribosomes are formed from two subunits in plants. The small subunit (40S) decodes messenger RNA, and the large subunit (60S) catalyzes peptide bond formation to form long chains of amino acids [44]. Several ribosomal proteins were upregulated in NA roots, implying that more protein synthesis sites were necessary for N-fixation plants. However, the majority of the protein translation initiation factors (11 proteins) were downregulated in NA roots, implying that the initiation step of protein synthesis was delayed. Meanwhile, the proteins involved in protein post-translational modification were also downregulated in NA roots, which participated in protein folding, protein degradation, methionine-(S)-S-oxide reductase activity, and protein phosphatase. Twelve ubiquitin proteins were upregulated in NA roots, implying that protein degradation was enhanced in N-fixation alfalfa roots. Therefore, *rhizobium* symbiosis remodified the protein synthesis and degradation process in the host plant root by modification of the ribosome protein abundance, decrease of translation initiation and post-translational modification, and enhancement of the protein degradation process, thus increasing protein metabolism.

### 4.2. Nodulation Triggered Pathways That Conferred Salt Stress in Alfalfa Roots

#### 4.2.1. Common Pathways for NA Roots under Control Conditions and Salt Stress

The pathways that were shared by NA roots under normal growth conditions and salt stress might be involved in improving salt tolerance, among which the upregulated pathways were biotic and abiotic stress, β-1,3-glucan hydrolases, glutathione S transferases, peroxidases, GDSL-motif lipase, ribosomal proteins, subtilisin-like proteases, and the downregulated pathways were FA synthesis, ethylene response, and redox. Those common pathways imply that stress response, signaling, energy metabolism, antioxidant defense, and protein synthesis and degradation were necessary for nitrogen fixation under salt stress.

#### 4.2.2. Unique Pathways under Salt Stress for NA Roots

NA roots also showed some unique pathways under salt stress, including cell wall degradation and modification, oxidases (copper and flavone), acid and other phosphatases, myrosinases-lectin-jacalin, plastocyanin-like, and porin transport.

Cell wall integrity plays a crucial role in plant growth and development, as well as response to biotic and abiotic stress [45]. The α-1,4-glucan-protein synthase proteins are involved in cellulose synthase, and three of them were upregulated in NA in a saline environment. Meanwhile, polygalacturonase-inhibitor proteins (PGIP) were greatly upregulated in NA roots. PGIP is a defense protein and prevents the degradation of pectin, which is a major cell wall component and plays important role in cell wall integrity [46]. Here, we identified three expansin proteins and two xyloglucan endotransglucosylase/hydrolase proteins (XTH) that were upregulated in NA roots. Expansins are cell wall-loosening enzymes, which mediate cell expansion and growth in under adverse environmental stresses [47]. XTH cleaves and reconnects xyloglucan molecules and is important for cell wall expansion and re-modeling [48]. Therefore, NA roots enhanced cell wall biosynthesis, attenuate cell wall degradation, and improved cell wall expansion, thus maintaining cell wall integrity and promoting cell growth in a saline environment.

Lectin domains are a major part of plant immune receptors, which play a crucial role in counteracting pathogen attacks [49]. Lectins bind to PAMPs (pathogen-associated molecular patterns) and DAMPs (damage-associated molecular patterns). These processes then trigger a downstream signaling cascade including ROS production, Ca^2+^ influx, and activation of the MAP kinase pathway. The signaling modified defense-related gene expression and defense molecule synthesis [50]. Four legume lectin β-domain proteins were highly up-regulated in NA roots by salt stress. Together with the upregulation of PGIP, those defense proteins might inhibit bacteria and fungus infections, enhance the innate immune system, and improve overall health status under salt stress. Alternatively, the defense proteins were increased to avoid excessive *rhizobium* infection and attenuated nitrogen fixation under salt stress, and therefore, save energy to survive a saline environment. This hypothesis was partially supported by the downregulation of plastocyanin-like proteins (PCs) in NA roots. PCs are blue copper-containing proteins and participated in rhizobia infection, nodules development, and nitrogenase activity [51].

The secretion of acid phosphatases (APases) is known as an effective strategy for surviving phosphate (Pi) deprivation conditions in plants [52]. The upregulation of four Apases might be a strategy for NA roots to improve P acquisition and relieve ionic toxicity caused by salt stress.

We also observed four porin/voltage-dependent anion-selective channel (VDAC) proteins that were upregulated in NA roots under salt stress. VDAC proteins regulate the transport of metabolites between mitochondria and the cytoplasm by switching between open and closed states in the outer mitochondrial membrane. The function of VDAC proteins is versatile, including plant growth and development, response to biotic and abiotic stress, and programmed cell death [53]. It was reported that the overexpression of *MsVDAC* from alfalfa improved the abiotic tolerance in tobacco [54]. Hence, the increase in VDAC abundance might play a role in improving salt stress in NA plants.

### 4.3. Alfalfa with or without Nodules Differently Responded to Salt Stress

Ubiquitin proteins were upregulated in both NA and NN roots, implying that the protein degradation process was enhanced under salt stress in alfalfa roots. However, despite protein degradation, alfalfa with or without nodules responded to salt stress by different pathways.

#### 4.3.1. Pathways That Participated in Salt Stress in NN Roots

The unique salt-response pathways for NN roots were defense response, amino acid metabolism, flavonoids, jasmonate synthesis, vitamin metabolism, carbohydrate metabolism, 14-3-3 signaling, storage proteins, and porin transports.

The defense response-related proteins were upregulated in NN roots under salt stress, including OPP pathway proteins and PR-proteins. Plant flavonoids regulated nodulatin gene expression and response to biotic and abiotic stress [55]. We found seven upregulated proteins that were involved in flavonoid synthesis, which were important in improving salt tolerance in NN roots. We also identified four upregulated transaminase proteins, implying that nitrogen metabolism was enhanced in NN roots in a saline environment. 14-3-3 proteins mediated salt stress response by selectively activating or inactivating SOS2 and PKS5 proteins to regulate plasm membrane (PM) Na^+^/H^+^ antiporter and PM H^+^-ATPase activities in a Ca^2+^-dependent manner [56]. The upregulated 14-3-3 proteins might act as a salt stress response signaling and increase the response to salt stress in NN roots.

However, the pathways of jasmonate synthesis, cell wall synthesis, and porin transports were downregulated in NN roots under salinity stress compared to control conditions. The low abundance of these proteins might partly explain that NN plants were more sensitive to salt stress than NA ones.

#### 4.3.2. Pathways That Participated in Salt Stress in NA Roots

The unique salt-response pathways for NA roots were lipid metabolism, abiotic stress, DNA synthesis, protein synthesis and folding, cell cytoskeleton, and archaeal/vacuolar H^+^-ATPase (V-ATPase).

Lipids are essential components of plant cell membranes. Remodeling membrane lipids is an important strategy for adaptation to adverse environmental conditions. Acyl-CoA-binding proteins (ACBPs) bind long-chain acyl-CoA esters with high specificity and affinity and are associated with plant stress response by regulating lipid metabolism [57]. The upregulation of ACBPs might confer plant tolerance to oxidative stress caused by high salt concentration. Ten heat shock proteins were upregulated in NA roots under NaCl stress. Abiotic stresses frequently result in protein aggregation and cause metabolic dysfunction. The main function of heat shock proteins (HSP) /chaperones is to act as a buffer to protect proteins from misfolding and aggregation [58]. Meanwhile, we also found that 12 protein folding-related proteins were upregulated in NA roots. Together with the HSP, these proteins help stabilize or ensure the correct folding of other proteins under salt conditions. The organization of microtubule networks is crucial for cell division, cell growth, and cell plate deposition [59]. The microtubule organization-related protein was upregulated in NA roots, which might be important for cell division and growth under salt stress.

The histones and 60S subunits were downregulated in NA roots, implying the pathways of DNA and protein synthesis were inhibited under salt stress. Interestingly, we found two archaeal/vacuolar-type H^+^-ATPases (subunit E) greatly downregulated in NA roots when responding to salt stress. The V-ATPase is an ATP-driven ion pump. The energy released by ATP hydrolysis is transformed into electrochemical potential gradients to regulate pH in the plant endomembrane system and is responsible for H^+^ translocation across membranes [60].

Contrary to the downregulation of V-ATPases in NA roots, a steady H^+^ influx was observed in NA roots under salt stress, implying that NA roots might engage other H^+^ pumps to regulate transmembrane pH gradients. Na^+^ interferes with K^+^ homeostasis and inhibits K^+^ influx under salt stress. The ability to maintain a balance of the K^+^/Na^+^ ratio is a key mechanism for plant salt tolerance [61]. The ability to maintain a stable low level of K^+^ efflux is important in improving salt tolerance in NA roots.

## 5. Conclusions

Our results suggested that *rhizobium* symbiosis improved the salt tolerance of alfalfa plants by mediating a series of molecular and physiological processes. Even though many proteins and pathways were identified, the mechanisms of nodulation regulating these processes were still unclear. Clarification of the regulatory mechanisms will be helpful in improving crop performance in a saline environment.

## Figures and Tables

**Figure 1 genes-13-02004-f001:**
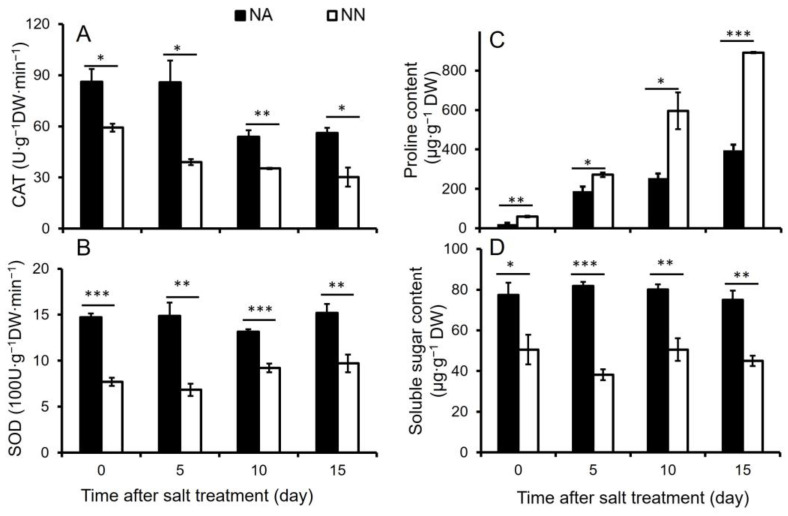
Effect of nodulation on catalase (CAT) activity (**A**), superoxide dismutase (SOD) activity (**B**), proline content (**C**), and soluble sugar content (**D**) in alfalfa roots under NaCl treatment. NA: alfalfa plants with active nodules; NN: alfalfa plants without nodules. Data are shown as means ± SE (*n* = 5). *, **, and *** indicate a significant difference between NA and NN roots at *p* < 0.05, *p* < 0.01, and *p* < 0.001, respectively.

**Figure 2 genes-13-02004-f002:**
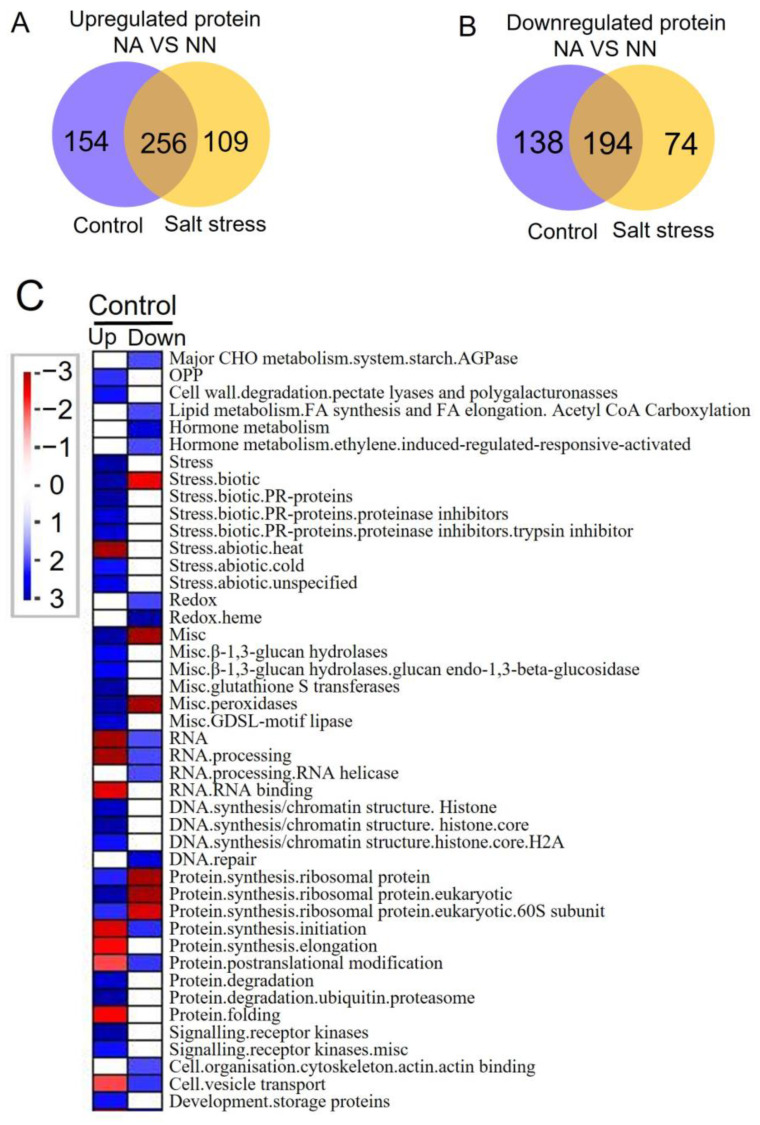
Venn diagrams and pathway enrichment of the differentially expressed proteins in comparison of NA and NN plants under normal growth conditions (control) and salt stress conditions. (**A**): the total 519 upregulated proteins; (**B**): the total 406 downregulated proteins; (**C**): the pathway changes when comparing NA and NN roots under control conditions with MapMan analysis; (**D**): the pathway changes when comparing NA and NN roots under salt stress conditions with MapMan analysis. Functional categories enrichment was made with Fisher’s exact red color: pathways were significantly under-represented; blue color: pathways are significantly over-represented. NA: alfalfa plants with active nodules; NN: alfalfa plants without nodules.

**Figure 3 genes-13-02004-f003:**
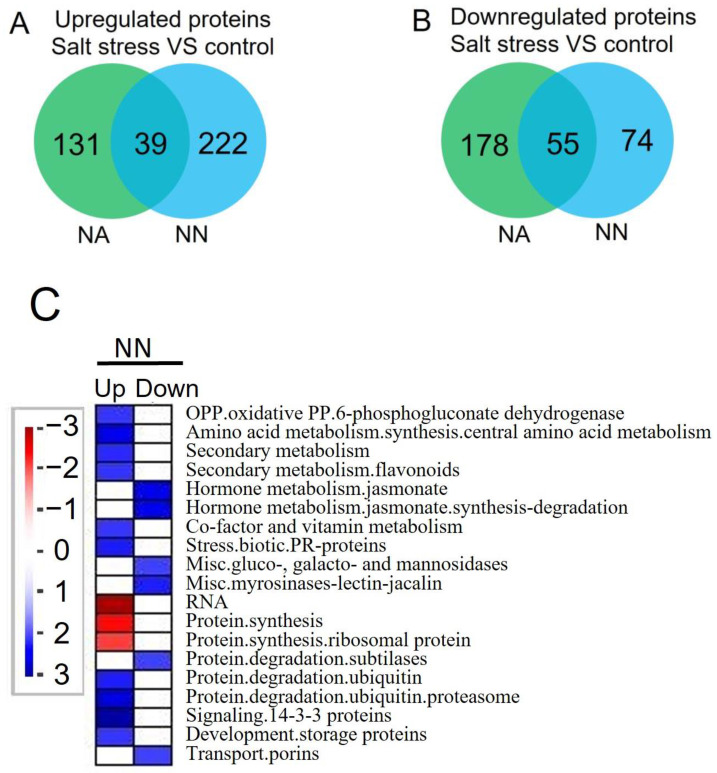
Venn diagrams and pathway enrichment of the DEPs in the roots in comparison of salt stress and normal conditions (control) for NA and NN roots. (**A**): the total 392 up-regulated proteins; (**B**): the total 307 down-regulated proteins; (**C**): the pathway changes of the NN plants when comparing salt stress and control conditions with MapMan analysis; (**D**): the pathway changes for NA plants when comparing salt stress and control conditions with MapMan analysis. Functional categories enrichment was made with Fisher’s exact. Blue color: pathways were significantly over-represented; red: pathways were significantly under-represented. NA: alfalfa plants with active nodules; NN: alfalfa plants without nodules.

**Figure 4 genes-13-02004-f004:**
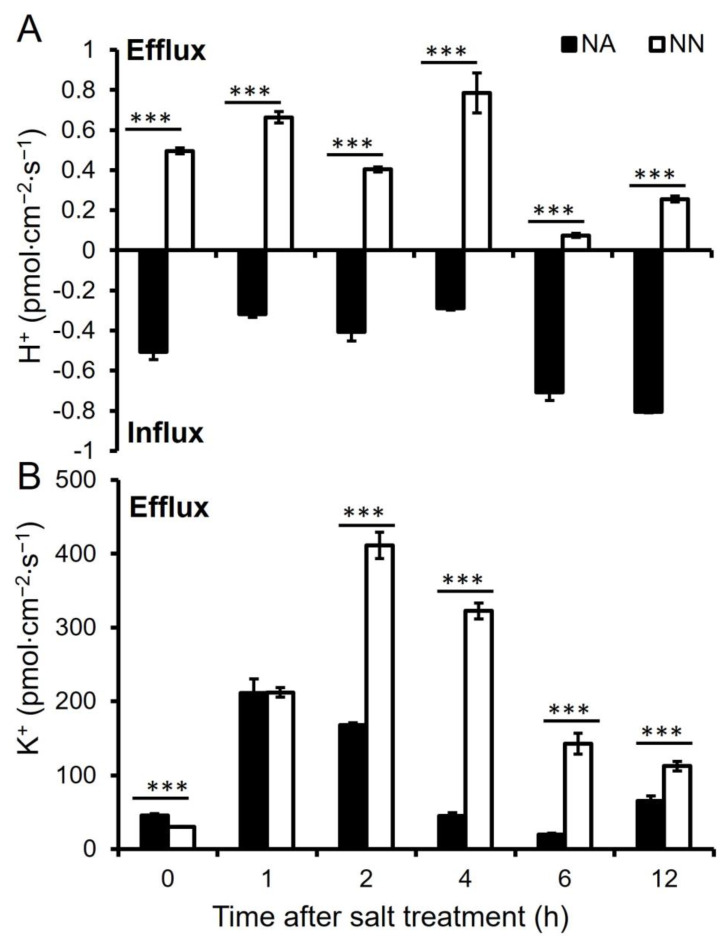
The net H^+^ (**A**) and K^+^ (**B**) flux under different periods of salt stress of alfalfa root tip. The positive and negative values indicate external and internal ion flux, respectively. NA: alfalfa plants with active nodules; NN: alfalfa plants without nodules. Data are shown as means ± SE (*n* = 8). *** indicates a significant difference between NA and NN roots at *p* < 0.001.

## Data Availability

Not applicable.

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
