# Peer review of "Proteomic Analysis of Alfalfa (Medicago sativa L.) Roots in Response to Rhizobium Nodulation and Salt Stress"

_genes, 2022, doi:10.3390/genes13112004_

Round 1

Reviewer 1 Report

The manuscript ID Genes - 1992219Proteomic Analysis of Alfalfa (Medicago sativa L.) Roots in Response to Rhizobium Nodulation and Salt Stress” presents an interesting topic. However, the paper is well written and requires minor revisions.

-          Line 15: Specify the pathways of biotic and abiotic stress, because there are already some mentioned such as antioxidant enzymes synthesis……etc

-          Figures 2 C, D and 3C, D – have insufficient quality; probably their resolution should be improved. Also change the text in these figures so that it can be read because it is too small and even if zoomed in the text is unclear.

-          Discussion, in all paragraphs; the figures and tables should not be cited in the discussion.

-          Several sections (for example lines 515 – 517) unnecessarily re-state the results, rather than focus on discussing the findings.

-         - Lines 567 – 584: This section requires revision to clearly outline the conclusions devoid of discussion, methodology, literature review. Conclusions should be drawn from the study and must be precise and linked to the objectives of the study without dragging in results and discussion.

Regards,

Author Response

Dear reviewers,

We would like to thank you for your valuable and constructive comments on our manuscript. We made revisions as suggested. The detailed responses are provided in the attachment.

Thank you very much for your help and we look forward to your final decision.

Sincerely,

Peizhi Yang, PhD

College of Grassland Agriculture

Northwest A&F University

Yangling, China

Reviewer 2 Report

Work by Wang et al. on "Proteomic Analysis of Alfalfa (Medicago sativa L.) Roots in Response to Rhizobium Nodulation and Salt Stress" is an extremely interesting literature, but requires several changes to increase its readability:

- please explain all abbreviations used in the work, e.g. IAA (line 40), NF (line 46) etc.

- the introduction section is too extensive, I propose to shorten it and clearly specify the research objective;

- please pay attention to whether the names of bacteria and plants are in the text everywhere in italics, e.g. line 105, 115 etc;

- please increase figure 1, because in its present state it is very difficult to read

- figure 2 is difficult to read in its present form, I propose to increase its resolution and size even over the entire manuscript page, as it presents extremely important data;

- the same remarks apply to figures 3 and 4, please increase the size and resolution of the presented data;

- minor editorial corrections: 4.1.5 cell cytoskeleton and 4.1.6 protein synthesis and degradation (please include a capital letter at the beginning of the section name); additional spaces in the text, e.g. line 118 etc.

Author Response

(The authors gave the same response as above.)
